# Multi-Level Deep Learning Model for Potato Leaf Disease Recognition

**Javed Rashid** [1,2,*][ID]**, Imran Khan** [1][ID]**, Ghulam Ali** [3][ID]**, Sultan H. Almotiri** [4][ID] **and Mohammed A. AlGhamdi** [4][ID] **and Khalid Masood** [5]

1   Department of CS&SE, Islamic International University, Islamabad 44000, Pakistan; imran.khan@iiu.edu.pk
2   Information Services, University of Okara, Okara 56310, Pakistan
3   Department of CS, University of Okara, Okara 56310, Pakistan; GhulamAli@uo.edu.pk
4   Computer Science Department, Umm Al-Qura University, Makkah 21961, Saudi Arabia;
    shmotiri@uqu.edu.sa (S.H.A.); maeghamdi@uqu.edu.sa (M.A.A.)
5   Department of Computer Science, Garrison University, Lahore 54000, Pakistan; kmasoodk@gmail.com
*   Correspondence: RanaJavedRashid@gmail.com; Tel.: +92-331-262-7786

**Abstract:** Potato leaf disease detection in an early stage is challenging because of variations in crop species, crop diseases symptoms and environmental factors. These factors make it difficult to detect potato leaf diseases in the early stage. Various machine learning techniques have been developed to detect potato leaf diseases. However, the existing methods cannot detect crop species and crop diseases in general because these models are trained and tested on images of plant leaves of a specific region. In this research, a multi-level deep learning model for potato leaf disease recognition has developed. At the first level, it extracts the potato leaves from the potato plant image using the YOLOv5 image segmentation technique. At the second level, a novel deep learning technique has been developed using a convolutional neural network to detect the early blight and late blight potato diseases from potato leaf images. The proposed potato leaf disease detection model was trained and tested on a potato leaf disease dataset. The potato leaf disease dataset contains 4062 images collected from the Central Punjab region of Pakistan. The proposed deep learning technique achieved 99.75% accuracy on the potato leaf disease dataset. The performance of the proposed techniques was also evaluated on the PlantVillage dataset. The proposed technique is also compared with the state-of-the-art models and achieved significantly concerning the accuracy and computational cost.

**Keywords:** potato leaf diseases; leaf disease detection; convolutional neural network; segmentation; early blight; late blight; deep learning

## 1. Introduction

The plant diseases affect the leaves, stems, roots and fruits; it also affects the crop quality and quantity, which causes food deprivation and insecurity throughout the world [1]. The estimated annual crop yield loss due to crop disease is about 16% globally, which is the primary cause of food shortage and increased food production costs [2]. According to the Food and Agriculture Organisation Report (FAO), the world's population will reach approximately 9.1 billion by 2050. For a steady food supply, about 70% of food production growth is required [3]. The factors affecting the plants and their products are categorised as diseases and disorders. The biotic factors are the diseases caused by algae, fungi, or bacteria, whereas the biotic factors inducing disorders are rainfall, moisture, temperature and nutrient deficiency [4].

There exist many methods to diagnose plant diseases; one of the primary and straightforward approaches is a visual estimation. The traditional plant disease diagnosing techniques depend on the farmer's experience, which is most uncertain and unreliable. Compared to the conventional plant disease diagnosing techniques, the researchers have introduced the spectrometer to diagnose the plant leaves as healthy and infected [5]. An-

other method is to extract the leaves' DNA by using the polymerase chain reaction [6] or real-time polymerase chain reaction [7]. Such techniques are challenging, expensive and time-consuming, require a highly professional operation, experiment condition and massive use of crop protection products. The recent advancement in Artificial Intelligence (AI), Machine Learning (ML) and Computer Vision (CV) technologies allow developing the automated plant leaf disease detection techniques. These techniques can efficiently and accurately detect plant leaf diseases in a brief time without human intervention. It has been observed that DL has been the most prominent usage in agriculture [8]. It helps to make substantial efforts to develop, control and enhance agricultural production.

Deep learning is the core of smart farming by adopting new devices, technologies, and algorithms in agriculture [9]. Deep learning is widely used to solve complex problems such as feature extraction, transformation, pattern analysis and image classification [10]. Many researchers used deep learning for crop disease diagnosing [11–13]. Chen et al. [11] proposed a deep learning model that counts the apples and oranges from the real-time images. Dias et al. [12] presented an apple flower semantic segmentation using the convolutional neural network (CNN) counting the number of flowers from the plants. Ubbens et al. [13] conducted a study to estimate the plant leaves using the CNN model.

Recently, numerous types of deep learning architecture has been proposed for plant disease classification. The most prominent technique is the convolutional neural network (CNN). A convolutional neural network is a supervised deep learning model inspired by the biological nervous system and vision system with significant performance compared to other models. As compared to Artificial Neural Network (ANN), CNN requires few neurons and multilayer convolution layers to learn the features, but it required an extensive dataset for training [3,10].

In the last few decades, several techniques have been developed to detect leaf diseases in various crops [8,14,15]. In most of the techniques, features were extracted using the image processing techniques, then extracted features were fed to a classification technique. Deepa and Nagarajan [16] proposed a plant leaf disease detection technique. The authors first applied the Kuan filter for noise removal and applied a Hough transformation to extract the colour, shape and texture features. A reweighted linear program boost classification was applied to classify the plant leaf disease. The PlantVillage dataset was used to evaluate the performance of the proposed technique. Karthik et al. [17] proposed a two-level deep learning technique for tomato leaf disease detection. The first model was applied to learn the significant features using residual learning, and the second-deep learning model was involved as an attention mechanism on top of the first model. The authors used the PlantVillage dataset to identify the late blight, early blight and leaf mold diseases of tomato crop. Zhang et al. [18] proposed an improved Faster RCNN method to determine the tomv, leaf mold fungus, blight and powdery mildew diseases of tomato crop. The researchers replaced the VGG16 model with a depth residual network to extract the features. For bounding boxes, a k-mean clustering algorithm was used. Sambasivam and Opiyo [19] researched cassava mosaic disease and cassava brown streak virus disease using convolutional neural networks. The dataset was imbalanced; therefore, training the model was challenging.

Many problems exist in the literature using deep learning approaches. The first problem is that the existing methods did not correctly identify the Pakistani region potato leaves diseases because all the current practices were trained on the PlantVillage dataset only. There is variation in potato diseases in different parts of the world due to variation in various factors such as shape, varieties and environmental factors. Therefore, the existing systems have a high false rate to recognise potato diseases in the Pakistani region. The second problem is that the PlantVillage dataset has fewer images, whereas to train any CNN model dataset should be huge. The plantVillage dataset has only 152 images for healthy potato leaf images. Suppose we split it into training, validation and testing by 80%, 10% and 10% ratios, respectively, then the normal leaf class has been further reduced for training. In that case, the existing methods have inadequate training in that class. The

plantVillage dataset has an imbalanced class as the late blight and early blight classes have 1000 images each; on the other hand, the normal leaf class has only 152 images. Then, there is a chance of over-fitting to train the model. Therefore, such a method failed to achieve high accuracy for the other regions of the world, such as Pakistan. Therefore, there is a dire need to develop a new dataset to detect the Pakistani region potato leaves' diseases in order for farmers in Pakistan can determine the diseases of potato in their early stage and enhance their income and boost the country's economy. The other problem is that most of the methods did not evaluate their performance on unseen images because the dataset was already minimal. The version of any model can be marked as good when it is tested on unseen data. Another problem is that the current methods have a low convergence speed due to the vast number of trainable parameters, and accuracy needs to be improved. The last problem in the literature is the non-availability of the potato leaf segmentation technique. This research is conducted to resolve the above research gaps.

The present research proposed a multi-level deep learning model for potato leaf disease recognition. At the first level, it extracts the potato leaves from the potato plant image using the YOLOv5 image segmentation technique. A novel potato leaf disease detection convolutional neural network (PDDCNN) has been developed at the second level to detect the early blight and late blight potato diseases from potato leaf images. Then, the performance of the proposed potato leaf disease detection convolutional neural network (PDDCNN) evaluated on the potato leaf dataset (PLD). The PLD dataset has been developed by capturing the images of potato leaves across various areas of Pakistan's Central Punjab region. The images are cropped and labelled with the help of plant pathologists.

The following are the main contributions of this research:

- A real-time novel Potato Leaf Segmentation and Extraction Technique using YOLOv5 has been developed to segment and extract potato leaves from the images.
- A novel deep learning technique called Potato Leaf Disease Detection using Convolutional Neural Network (PDDCNN) has been developed to detect the early blight, late blight diseases from potato leaf images.
- The proposed method has an optimal number of parameters as compared to state-of-the-art models.
- The development of a potato leaf disease dataset from the Central Punjab region of Pakistan by capturing three types of potato leaf images: early blight, late blight and healthy.

The rest of the article is organised as related work is presented in Section 2, materials and methods is in Section 3, results and discussion are described in Section 4, while Section 5 presents the conclusion and future work followed by the references.

## 2. Related Work

In the last few decades, many researchers worked on multiple crops, including potatoes; their focus was not on the single potato crop diseases [20–22]. The models were trained on specific region dataset (PlantVillage [23]), which was developed in the USA and Switzerland. The diseases of potato vary from other regions due to the difference in leaf shapes, varieties and environmental factors [24]. Geetharamani and Pandian [20] proposed a deep CNN model to differentiate between healthy and unhealthy leaves of multiple crops. The model was trained using the PlantVillage dataset with 38 different types of crops with disease leaf images, healthy leaf images and background images. The focus of the model was not on single potato crop diseases. The model is also trained in specific region dataset USA and Switzerland, which failed to detect the Pakistani region potato leaf diseases. Kamal et al. [21] developed plant leaf disease identification models named Modified MobileNet and Reduced MobileNet using depthwise separable convolution instead of convolution layer by modifying the MobileNet [25]. The proposed model was trained on multiple crops of the PlantVillage dataset, where the plant leaf images were collected from a specific region of the world. Khamparia et al. in [22], proposed a hybrid approach to detect crop leaf disease using the combination of CNN and autoencoders. The

model was trained on the PlantVillage dataset for multiple crop diseases and specific region diseases. In [26], Liang et al. proposed a plant disease diagnosis and severity estimation network based on a residual structure and shuffle units of ResNet50 architecture [27]. The PlantVillage dataset was also used to detect the multiple crop diseases of a specific region. Ferentinos [28] investigated AlexNet [28], Overfeat [29], AlexNetOWTBn [30], VGG [31] and GoogLeNet [32] deep learning-based architectures in order to identify the normal or abnormal plants from plant leaf images. The researchers performed the transfer learning approach using the PlantVillage dataset to detect the specific region's multiple crops diseases.

Many researchers worked on potato crops diseases but also trained the models on a specific dataset PlanVillage. Khalifa et al. [33] proposed a CNN model to detect early blight and late blight diseases along with a healthy class. The researchers trained their model on the PlantVillage dataset, which is for specific regions' crops only. Rozaqi and Sunyoto [34] proposed a CNN model to detect the early blight, late blight disease of potato, and a healthy class. They trained the model on the PlantVillage dataset to detect the diseases of a specific region. Sanjeev et al. [35] proposed a Feed-Forward Neural Network (FFNN) to detect early blight, late blight diseases along with healthy leaves. The proposed method was trained and tested on the PlantVillage dataset. Barman et al. [36] proposed a self-build CNN (SBCNN) model to detect the early blight, late blight potato leaf diseases, and healthy class. The PlantVillage dataset was also used to train the model, which is for a specific region. They did not validate their model on unseen test data. Tiwari et al. [37] used a pre-trained model VGG19 to extract the features and used multiple classifiers KNN, SVM and neural network for classification. The model also trained on the PlantVillage dataset to detect the early blight and late blight disease of potato leaves. They did not test their model on unseen data. Lee et al. [38] developed a CNN model to detect the early blight, late blight diseases, and healthy leaves of potato. The researchers also used the PlantVillage dataset belonging to a specific region. The model was not tested on unseen data. Islam et al. [39] proposed a segment-based and multi-SVM-based model to detect potato diseases, such as early blight, late blight and healthy leaves. Their method also used the PlantVillage dataset and also needs to be improved in terms of accuracy. As shown in Table 1.

**Table 1.** Summary of related work.

| Reference | Methodology | Plant Name | Disease | Dataset | Accuracy |
|-----------|-------------|------------|---------|---------|----------|
| [16] | Reweighted Linear Boost Program Classification | Multiple | Multiple | PlantVillage | 92% |
| [17] | Attention based Residual Network | Tomato | Early Blight, Late Blight Leaf Mold | PlantVillage | 98% |
| [18] | Faster RCNN | Tomato | Blight, Powdery Mildew, ToMV, Leaf Mold Fungus | AIChallenger | 97.1% |
| [19] | CNNs | Cassava | Cassava Mosaic, Cassava Brown Streak Virus | Cassava Challenge | 93% |
| [20] | Deep CNN | Multiple (Potato) | Multiple | PlantVillage | 96.46% |
| [21] | Modified MobileNet | Multiple (Potato) | Multiple | PlantVillage | 98.34% |
| [22] | CNN and Autoencoders | Potato, Maize, Tomato | Multiple | PlantVillage | 97.50% 100% |
| [26] | ResNet50 | Multiple (Potato) | Multiple | PlantVillage | 98% |
| [28] | AlexNet, Overfeat AlexNetOWTBn, VGG and GoogLeNet | Multiple (Potato) | Multiple | PlantVillage | 99.53% |
| [33] | CNN | Potato | Early Blight, Late Blight | PlantVillage | 98% |

**Table 1.** *Cont.*

| Reference | Methodology | Plant Name | Disease | Dataset | Accuracy |
|---|---|---|---|---|---|
| [34] | CNN | Potato | Early Blight, Late Blight | PlantVillage | 92% |
| [35] | FFNN | Potato | Early Blight, Late Blight | PlantVillage | 96.5% |
| [36] | SBCNN | Potato | Early Blight, Late Blight | PlantVillage | 96.75% |
| [37] | SVM, KNN and Neural Net | Potato | Early Blight, Late Blight | PlantVillage | 97.8% |
| [38] | CNN | Potato | Early Blight, Late Blight | PlantVillage | 99% |
| [39] | Segment and Multi SVM | Potato | Early Blight, Late Blight | PlantVillage | 95% |

## 3. Materials and Methods

Many problems exist in the literature using deep learning approaches, including incorrect identification of potato leaf diseases, variation in potato diseases, varieties and environmental factors. The existing systems have a high false rate to recognise potato diseases in the Pakistani region. The existing potato leaves disease datasets contain inadequate training samples with imbalanced class samples. Another problem is that the current methods have a low convergence speed due to the vast number of trainable parameters, and accuracy needs to be improved. The last problem in the literature is the non-availability of the potato leaf segmentation technique. A multi-level deep learning model for potato leaf disease recognition is proposed to classify the potato leaves diseases in this research. At the first level, it extracts the potato leaves from the potato plant image using the YOLOv5 image segmentation technique. A novel potato leaf disease detection convolutional neural network (PDDCNN) has been developed at the second level to detect the early blight and late blight potato diseases from potato leaf images. The flow chart of the proposed method is shown in Figure 1, the algorithm is described in Algorithm 1, and the proposed methodology overall architecture is shown in Figure 2.

---

**Algorithm 1** Multi-Level Deep Learning Model for Potato Leaf Disease Recognition Algorithm

---

1. Capture the real-time videos and images of Potato plants from the lab and field environment.
2. Convert the videos to frames (images).
3. Annotate the potato leaf images (single class) and save the annotation in YOLOv5 and XML format.
4. Divide the Potato Leaf Images Dataset into training, validation and testing sets.
5. Pre-processing (auto-orient and resize) is applied to the annotated images.
6. Pre-processing (data augmentation) is applied to the training set.
7. Save the Potato Leaf Images Dataset into YOLOv5 PyTorch format.
8. Upload the dataset into google drive.
9. Train and validate the custom YOLOv5s model with the help of Google Colab by using the Potato Leaf Images Dataset.
10. Classification output of YoloV5s model and using the annotations of Potato Leaf Images Dataset, potato leaves were extracted/segmented and made Potato Leaf Disease Dataset (PLD).
11. Label the images of PLD with the help of plant pathologists with their respective classes.
12. Pre-process all the images by applying data augmentation.
13. Divide the dataset among training, validation and testing with 80%, 10% and 10% ratios, respectively.
14. Train the CNN model with the help of training images.
15. Use the validation images to validate the CNN model at the end of each epoch.
16. Save the Trained PDDCNN Model.
17. Testing is applied to the PDDCNN trained model using testing images.

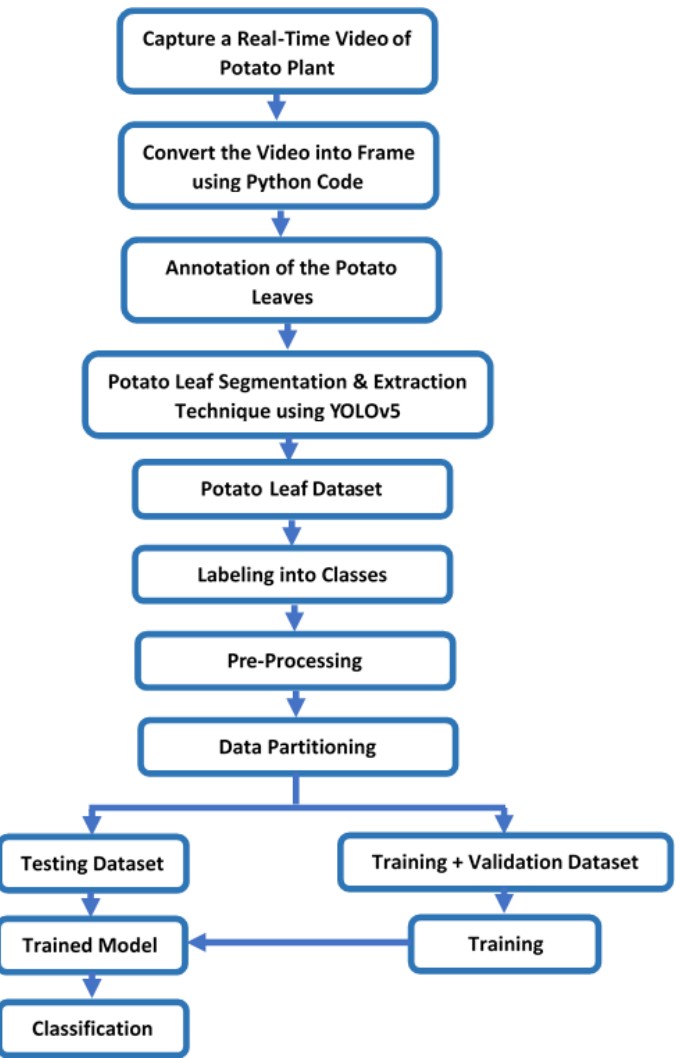

**Figure 1.** A flowchart of the proposed model.

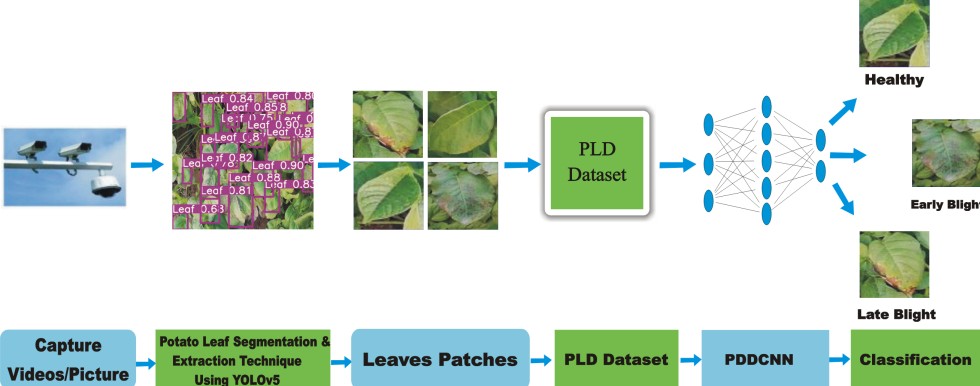

**Figure 2.** The overall architecture of the proposed model.

*3.1. Dataset*

The performance of deep learning models heavily depends upon an appropriate and valid dataset. In this research, the following datasets are used.

3.1.1. PlantVillage Dataset

The PDDCNN method's performance is assessed using the potato leaf images of a publicly available dataset called PlantVillage [23]. The PlantVillage dataset was developed

by Penn State University (US) and EPFL (Switzerland), which is a non-profit project. The database consists of JPG colour images with $256 \times 256$ dimensions. It has 38 classes of diseased and healthy leaves of 14 plants. The focus of this research is on the potato crop. Therefore, 1000 leaves for late blight, 1000 leaves for early blight, and 152 images of healthy leaves were selected for the experimental purposes, as shown in Table 2.

**Table 2.** Summary of the PlantVillage dataset.

| PlantVillage Dataset | |
|---|---|
| **Class Labels** | **Samples** |
| Early Blight | 1000 |
| Late Blight | 1000 |
| Healthy | 152 |
| **Total Samples** | **2152** |

3.1.2. Potato Disease Leaf Dataset

In the literature, only the PlantVillage dataset has been used to develop the models because only the PlantVillage dataset is publicly available for potato leaf diseases. All the researchers used the PlantVillage dataset in their research, but there are many research gaps found in the literature. The PlantVillage dataset has been developed from the specific region under particular geography and environmental factors. There is variation in potato diseases of different parts of the world due to variation in various factors such as shape, varieties and environmental factors. Therefore, the existing systems have a high false rate to recognize potato disease detection in the Pakistani region potato leaf images, as shown in Table 3. The PlantVillage dataset also has fewer images and an imbalanced class distribution. Therefore, there is a dire need to develop a new potato leaves dataset collecting from the Pakistani areas. It will help the researchers train their models to identify Pakistan's potato leaf diseases that will be useful for Pakistani farmers to detect the potato diseases in their early stage.

**Table 3.** Classification Accuracies of the proposed PDDCNN model training on PlantVillage and testing on the PLD dataset.

| Training Dataset | Testing Dataset | Early Blight Accuracy | Healthy Accuracy | Late Blight Accuracy | Total Images | Overall Testing Accuracy |
|---|---|---|---|---|---|---|
| PlantVillage | PLD | 95.71% | 08.82% | 23.94% | 807 | 48.89% |

Thus, a new Potato Leaf Dataset (PLD) has been developed from Pakistan's Central Punjab region. We collected our real-time dataset in the form of videos and pictures. Different capturing devices, such as mobile phone cameras, digital cameras and drones, were used to make the variations in the real-time dataset. The capturing distance for the mobile phone cameras and digital cameras were 1–2 feet, whereas the capturing distance for the drone was set at 5–10 feet. Drone fanning distorted the videos and images because of plant leaves movement; therefore, we maximised the plant and drone distance as much as possible. We selected the district Okara in the Central Punjab region of Pakistan due to the higher cultivation of potato and focus on the varieties of potato found in the district Okara: Coroda, Mozika and Sante. Potatoes of different varieties acclimatised to the native environment were sown in the agricultural land exposed to sunny conditions during November, 2020. Potatoes were grown in rows and segregated at a distance of 3 feet apart from each other. Seeds of the plants were cultivated by digging the soil pit hole to a depth of 6–8 inches and having a 5-inch width. Seeds were placed in the pit hole and were covered with the manure mixed soil and further irrigated with canal water. We captured the images and videos with varying conditions, i.e., morning, evening, noon, cloudy, sunny, rainy, etc. The healthy and infected leaves were annotated with the use of the LabelMe tool into YOLOv5 PyTorch format and XML format. For segmentation and leaf extraction,

the YOLOv5s model is trained from scratch. With the help of YOLOv5s model output and annotations, potato leaves were extracted with the help of Python code. With the help of plant pathologists, a total of 4062 potato healthy and diseased leaf images were selected in the PLD dataset. Then, plant pathologists labelled the images into early blight, late blight and normal leaf classes. The plant leaf dataset consisted of 1628, 1414 and 1020 potato leaf images for early blight, late blight and healthy classes, respectively, as described in Table 4. The sample images of the PLD dataset are shown in Figure 3. The PLD dataset can be accessed from https://drive.google.com/drive/folders/1FpcQA66pEg0XR8y5uEzWU_ _REPpqSAPD?usp=sharing, accessed on 20 June 2021.

**Table 4.** Summary of PLD datasets.

| Potato Leaf Dataset (PLD) | |
|---|---|
| **Class Labels** | **Samples** |
| Early Blight | 1628 |
| Late Blight | 1414 |
| Healthy | 1020 |
| **Total Samples** | **4062** |

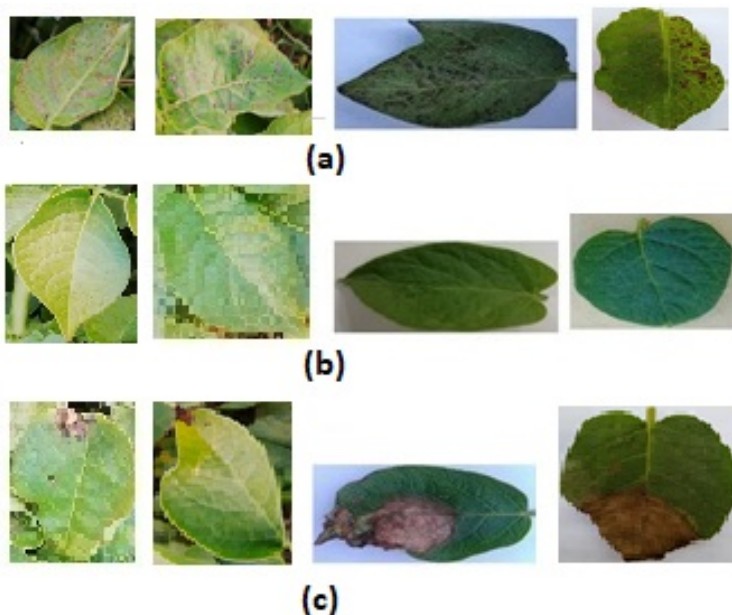

**Figure 3.** Examples of potato leaf images: (**a**) early blight, (**b**) healthy and (**c**) late blight.

### 3.2. Image Pre-Processing

For achieving more consistency in classification results and better feature extraction, pre-processing was applied to the final images of the PLD. The CNN method needed a lot of iterative training; for this purpose, a large-scale image dataset was required to eliminate the chance of overfitting.

### 3.2.1. Data Augmentation

Different data augmentation techniques were applied to the training set using the Image Data Generator method of Keras library in Python to overcome overfitting and enhance the dataset's diversity. The computational cost was reduced using the smaller pixel values and the same range; for this purpose, it used scale transformation. Therefore, every pixel value was ranged from 0 to 1 using the parameter value (1./255). Images were rotated to a specific angle using the rotation transformation; therefore, 25° was employed to rotate the images. Images can shift randomly either towards the right or left by using the width shift range transformation; selected a 0.1 value of the width shift parameter. Training

images moved vertically using the height shift range parameter with a 0.1 range value. In the shear transformation, one axis was fixed of the image and then stretched the other axis to a specific angle known as the shear angle; therefore, a 0.2 shear angle was applied. The zoom range argument was applied to perform the random zoom transformation; >1.0 means magnifying the images, and <1.0 was used to zoom out the image; therefore, 0.2 zoom_range was employed to magnify the image. Flip was applied to flip the image horizontally. Brightness transformation was applied, in which 0.0 means no brightness, and 1.0 means maximum brightness; therefore, we employed a 0.5–1.0 zoom range. In channel shift transformation, randomly shift the channel values by a random value was selected from the specified range; therefore, a 0.05 channel shift range was applied, and the fill mode was nearest.

### 3.3. Training, Validation and Testing

The entire PLD dataset was divided into three parts, training, validation and testing. The training dataset was used to train the PDDCNN model, while we utilised the validation and test dataset to evaluate the proposed model's performance. Therefore, we split the training, validation and testing datasets with 80%, 10% and 10%, respectively. For the PLD dataset, 3257, 403 and 403 images for training, validation and testing were used, respectively. Different data augmentation techniques performed on the training set, i.e., rescaling, rotation, width shift, height shift, shear range, zoom range, horizontal flip, brightness and channel shift with the fill mode nearest to increase the diversity and enhance the dataset. It would overcome the overfitting problem, thus ensuring the generalisation of the model.

In the CNN model, training was performed on the training samples from the input layer to the output layer, making a prediction, and errors or results were figured out. In the case of a wrong prediction, back-propagation was performed in reverse order. Therefore, in the current research, the back-propagation algorithm was applied to adjust the model weights accordingly for a better prognosis. The complete process of forwarding and back-propagation was known as one epoch. The model used the Adam optimising algorithm for the research. The current study had taken the training images from class labels early blight, healthy and late blight, respectively, while maintaining the 80% image ratios. The remaining 20% of untouched images were further split into validation and testing with a 10% ratio each on both datasets. The proposed PDDCNN model was trained on a training dataset to classify and predict every training image's class label.

### 3.4. Potato Leaf Segmentation and Extraction Technique Using YOLOv5

The latest product of the YOLO architecture series is the YOLOv5 network [40,41]. The recognition exactness of this organisation model is high, and the inference speed is quick, with the quickest identification speed being 140 frames each second. Then again, the size of the weight file of YOLOv5 target identification network model is small, which is almost 90% more modest than YOLOv4, demonstrating that YOLOv5 model is appropriate for deployment to the embedded devices to implement instantaneous detection. Hence, the benefits of YOLOv5 network are its high detection accuracy, lightweight attributes and quick recognition speed simultaneously. The YOLOv5 architecture comprehends four architectures, specifically named YOLOv5l [41], YOLOv5x [41], YOLOv5m [41] and YOLOv5s [41], correspondingly. The key modification between them is that the amount of feature extraction modules and convolution kernel in the specific location of the network is diverse. The number of model parameters and the size of models in the four architectures increase in turn. In this research, we used YOLOv5s architecture, as shown in Figure 4.

The YOLOv5s [41] framework primarily comprises three elements, including neck network, backbone network and detect network. A backbone network is a convolutional neural network (CNN) that combines diverse fine-grained images and forms image features. Precisely, the first layer of the backbone is intended to decrease the calculation of the model and speed up the training speed. Its functions are as follows: Initially, the input 3 channel

image (the default input image size of YOLOv5s architecture is $3 \times 640 \times 640$) was segmented into four portions with the size of $3 \times 320 \times 320$ per slice, using a slicing procedure. Furthermore, the concat procedure was applied to connect the four portions in-depth, with the size of the output feature map being $12 \times 320 \times 320$, and then through the convolutional layer composed of 32 convolution kernels, the output feature map with a size of $32 \times 320 \times 320$ was produced. The outcomes were output into the next layer to conclude through the BN layer (batch normalisation) and the Hardswish activation functions. BottleneckCSP module is the third layer of the backbone network, which is intended to extract the deep features of the image better. The BottleneckCSP is primarily composed of a Bottleneck module, a residual network architecture that joins a convolutional layer (Conv2d + BN + Hardswish activation function) with a convolution kernel size of $1 \times 1$ and kernel size of $3 \times 3$. The final output of the Bottleneck module is the addition of the output of this part and the initial input through the residual structure. BottleneckCSP module initial input is input into two divisions, and the volume of channels of feature maps is halved through the convolution operation in two divisions. Simultaneously, through the Conv2d layer and Bottleneck module in branch two, the output feature map of branch one and two are linked in-depth using the concat operation. Finally, the output feature map of the module was achieved after passing through the Conv2d layer and Batch Normalisation (BN) layer sequentially, and the size of this feature map and input of the BottleneckCSP module is the same.

The SPP module (spatial pyramid pooling) is the ninth layer of the Backbone network, which is intended to recover the receptive field of the network by converting any size of the feature map into a fixed-size feature vector. The size of the input feature map of the SPP module belonged to YOLOv5s is $512 \times 20 \times 20$. Initially, the feature map with a size of $256 \times 20 \times 20$ is output after a pass through the convolutional layer; the convolution kernel size is $1 \times 1$. Formerly, this feature map and the output feature map that are subsampled through three parallel maxpooling layers are connected in-depth. The size of the output feature map is $1024 \times 20 \times 20$. Lastly, the final output feature map with a $512 \times 20 \times 20$ is obtained after a pass through the convolutional layer with a 512 convolution kernel. The neck network is a series of feature aggregation layers of mixed and combined image features, primarily utilised to generate FPN (feature pyramid networks). Then the output feature map is conveyed to the detect network (prediction network). Meanwhile, the feature extractor of this network adopts a new FPN structure, which improves the bottom-up path, the transmission of low-level features and the recognition of objects with different scales. Therefore, the same target object with different sizes and scales can be precisely identified.

The recognition network is primarily utilised for the final recognition part of the model, which relates anchor boxes on the feature map output from the previous layer. It outputs a vector with the category probability of the target object, the object score and the position of the bounding box surrounding the object. The recognition network of YOLOv5s architecture comprises three detect layers, whose input is a feature map with dimensions of $80 \times 80$, $40 \times 40$ and $20 \times 20$ correspondingly, utilised to identify the image objects of different sizes. Each detect layer finally outputs a 21-channel vector ((2 classes + 1 class probability + 4 surrounding box position coordinates) $\times$ 3 anchor boxes). Then the expected bounding boxes and class of the targets in the original image were produced and labelled, applying the recognition of the leaves in the image.

YOLOv5s model was trained from scratch with default hyperparameters, and 100 epochs were used with image size $416 \times 416$, batch size 32. First, the output of the trained model is stored into a YOLOv5 format file, then this text file and annotation stored in files were stored in a CSV file. Then using the python code, the annotations of the leaves were cropped and stored in a folder in jpg image format.

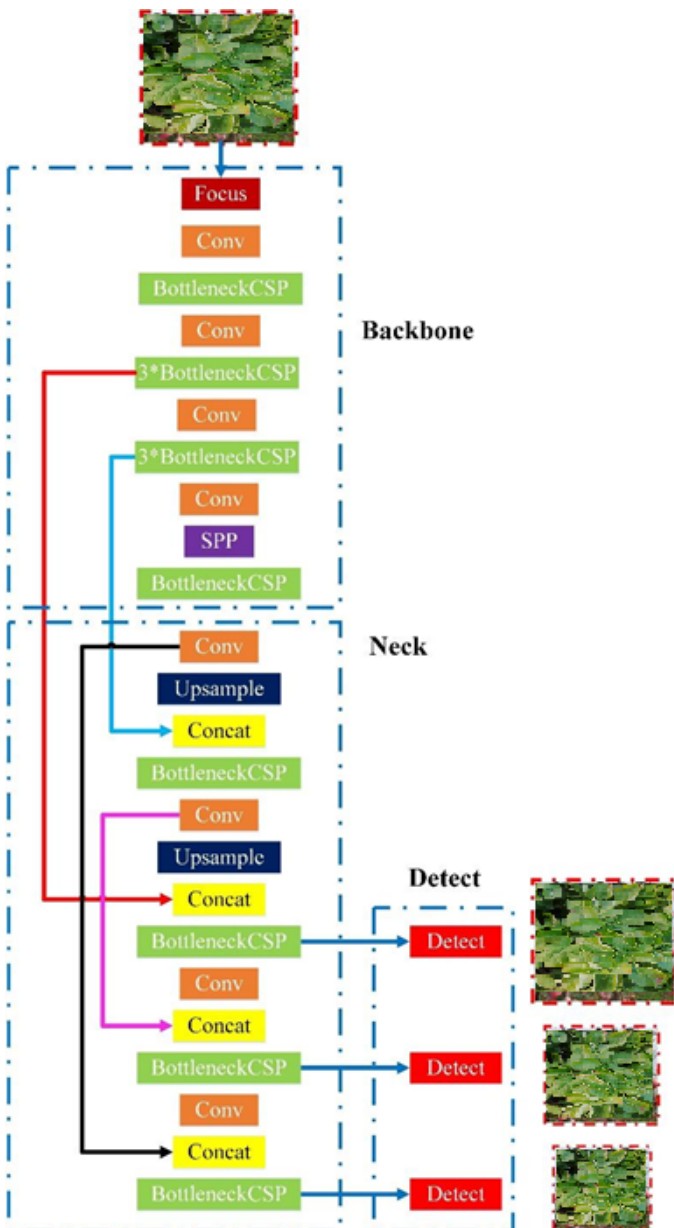

**Figure 4.** YOLOv5s model architecture.

*3.5. Potato Leaf Disease Detection Using Convolutional Neural Network (PDDCNN)*

Applications related to deep learning (DL) have emerged with the technological advancement in efficient computational devices, such as Graphics Processing Unit (GPU). The concept of DL was motivated by the conventional artificial neural network. In deep learning, CNN played a vital role in which many preprocessing layers were stacked to extract the essential features. These features were fed into fully connected layers for final decision. DL models had massively developed after Krizhevsky et al. [42] achieved tremendous image classification accuracy on CNN in 2012. Since then, CNN had applied in many DL applications, i.e., pattern recognition, image classification, object detection, voice recognition, etc. [43,44].

Figure 5 exhibited the architecture of the proposed PDDCNN used to classify the potato leaf disease along with healthy leaves. The model consisted of three convolutional layers, where each layer was followed by Rectified Linear Unit (ReLu) and max-pooling layers. It used the flatten layer to convert the convolved matrix into a 1D array. After flattening, the model used four dense or fully connected layers. First, three fully connected layers used the activation function ReLu. The last fully connected layer, or the output layer,

used the activation function Softmax because it was a multiclass model. In this research, we used Adam optimiser and categorial_cross_entropy loss function.

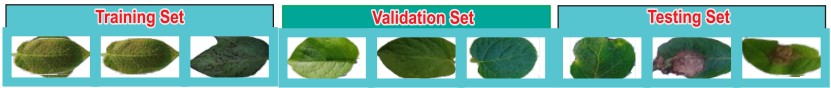

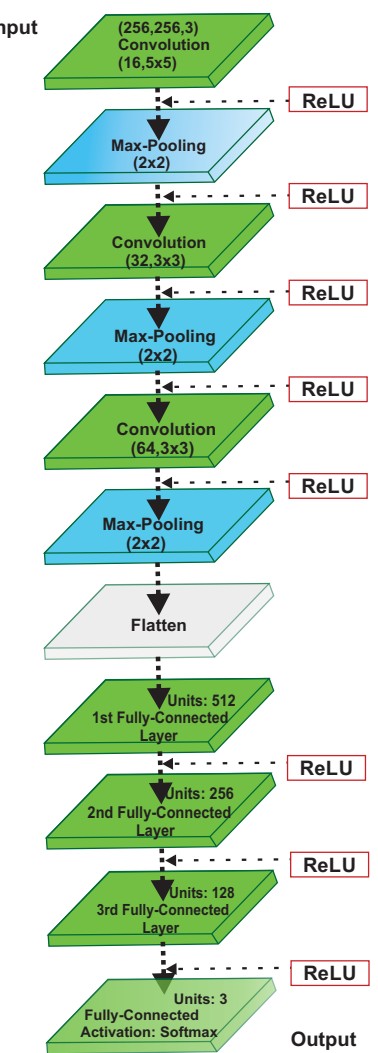

**Figure 5.** The architecture of the proposed PDDCNN model.

In the convolutional process, the input volume was convolved with the weights. The convolved matrix might be shrunk or expanded depending on the stride and padding. The convolutional process reduced the spatial height and width, but depth was increasing. Each convolutional layer applied the ReLu nonlinear action function, which converted the negative values to zero and reduced the vanishing gradient probability. It used pooling to reduce the computational cost and spatial size. Max pooling was applied to downsample the images, which reduced the overfitting and improved the activation function's performance, thus improving the convergence speed. A fully connected or dense layer was the final output layer responsible for predicting a potato leaf image class.

Further details of the PDDCNN are given below:

1. The sequential model used a series of layers to extract the input image's essential features for further processing.

2. The first convolution layer with input image shape was 256 × 256 × 3, 16 filters, kernel size 3 × 3, padding with stride 1 and activation was ReLu.
3. Image size was reduced using the max-pooling layer with pool size (2,2) after the first convolutional layer.
4. The second convolutional layer used 32 filters with kernel size 3 × 3, stride value 1 and the ReLu nonlinear activation function.
5. After the second convolutional layer, max-pooling was applied with pool size (2,2).
6. Then third and last convolutional layer used 64 filters with kernel size 3 × 3. The stride was 1 and, again, used the padding and activation function ReLu.
7. Then, converted the convolved matrix into a 1D vector using the flatten layer.
8. It used the four hidden or fully connected layers for classification or decision-making based on generated features.
9. The first fully connected layer or dense or hidden layer was used with 512 neurons, followed by ReLu activation functions.
10. The second fully connected layer or dense or hidden layer was used with 256 neurons, followed by ReLu activation functions.
11. It used 128 neurons and the ReLu activation function as a third hidden layer.
12. The output neurons always depended on the number of classes. The current research was a multi-classification problem with three classes; therefore, the last hidden or output layer used three neurons and a softmax activation function.
13. The overall accuracy of the model was evaluated by predicting the class label in the output layer.

The configuration details and various parameters of the proposed PDDCNN are given in Table 5.

**Table 5.** The proposed PDDCNN model configuration details of various parameters.

| | |
|---|---|
| Convolution Layers | 3 (with 3 × 3 filters/kernels each) |
| Max-Pooling Layers | 3 (with (2, 2) pool size each) |
| Hidden Layer Neurons | 512 (1st), 256 (2nd), 128 (3rd) |
| Output Layer Activation Function | Softmax |
| Batch size | 32 |
| Epochs | 100 |
| Training Optimiser | Adam |
| Loss Function | Categorial Cross Entropy |

*3.6. Evaluation Measures*

3.6.1. Classification Accuracy

Classification accuracy is calculated by the number of correct predictions divided by the total number of accurate predictions.

$$Accuracy = \frac{Number\ of\ Correct\ Predictions}{Total\ Number\ of\ Predictions}$$

3.6.2. Precision

There are numerous cases in which classification accuracy is not a significant pointer to measure the model's performance. One of these scenarios is when class dissemination is imbalanced. If you anticipate all samples as the top class, you will get a high accuracy rate, which does not make sense (since the model is not learning anything, and it is fair foreseeing everything as the best class. Subsequently, precision describes the inconsistency you find when using the same instrument; you repeatedly measure the same part. Precision is one of such measures, which is characterised as:

$$Precision = \frac{TP}{(TP + FP)}$$

### 3.6.3. Recall

The recall is another critical metric, characterised as the division of input samples from a class accurately anticipated by the model. The recall is calculated as:

$$Recall = \frac{TP}{(TP + FN)}$$

### 3.6.4. F1 Score

One well-known metric that combines precision and recall is called the F1-score, which is defined as:

$$F1Score = \frac{2 * Precision * Recall}{(Precision + Recall)}$$

### 3.6.5. ROC Curve

The receiver operating characteristic curve (ROC) is a plot that appears as the execution of a classifier as work of its cutoff limit. It seems the TPR against the FPR for different limit values. ROC curve could be a well-known curve to demonstrate performance and choose an excellent model's excellent cutoff threshold.

## 4. Results and Discussion

The proposed PDDCNN method experiments were implemented using the Tensor-Flow framework [29], Keras open source libraries and Python programming language. It utilised the Adam optimiser with a default learning rate and categorical cross-entropy loss function for training. The proposed PDDCNN model experiments were conducted on Google Colab.

The results of the proposed PDDCNN model focused on:

1.  Differentiating the potato leaf images into early blight, late blight, or healthy.
2.  The evaluation of the proposed PDDCNN model's performance on the PLD dataset using data augmentation and without data augmentation techniques on the training set.
3.  An assessment of the proposed PDDCNN model's performance on the publicly available dataset PlantVillage by applying data augmentation and without data augmentation techniques.
4.  We measured the performance of the proposed model on the cross dataset.
5.  To compare the results with other state-of-the-art networks, such as VGG16 [31], InceptionResNetV2 [45], DenseNet_121 [46], DenseNet169 [46] and Xception [47], using the transfer learning.
6.  To evaluate the results of potato leaf disease detection with existing studies using deep learning.

### 4.1. Proposed PDDCNN Model Performance on PLD Dataset

Two set of experiments were conducted to evaluate the performance of the proposed PDDCNN model. In the first set of experiments, we applied four groups (sets) of data augmentation techniques to the PLD dataset's training set. In the second experiment, we performed training without using the data augmentation techniques. All experiments used the Adam optimiser, categorial-cross-entropy loss function, 32 batch size, 100 epochs and the default learning rate. The comparison of four sets of data augmentation techniques applied to the PLD dataset using the values of the parameters in Section 3.2.1, the results of all groups shown in Table 6. Set #1 used only one data augmentation technique and achieved 97.56% accuracy; the set #2 used two data augmentation techniques and gained 98.28% accuracy. Set #3 achieved 99.02% accuracy with five data augmentation techniques and had 99.75% accuracy using seven data augmentation techniques. It confirmed that as we increased the training samples using more data augmentation techniques, the accuracy also increased. The set # achieved the highest accuracy because we increased the training

samples using the seven data augmentation techniques. The results showed that the PDDCNN required a vast amount of training samples for training.

**Table 6.** Classification accuracies on different sets of data augmentations of the proposed PDDCNN model on the PLD dataset.

| Set # | Data Augmentation Used | Early Blight | Healthy | Late Blight | Average |
|---|---|---|---|---|---|
| 1 | rotation_range | 96.32% | 97.06% | 96.48% | 97.56% |
| 2 | rotation_range, width_shift_range, height_shift_range | 98.77% | 98.04% | 97.89% | 98.28% |
| 3 | width_shift_range, height_shift_range, shear_range, zoom_range, horizontal_flip | 99.39% | 99.02% | 98.59% | 99.02% |
| 4 | rotation_range, width_shift_range, height_shift_range, shear_range, zoom_range, horizontal_flip, brightness_range, channel_shift_range, fill_mode = nearest | 99.38% | 100% | 100% | 99.75% |

Table 7 showed the results achieved in data augmentation techniques achieved in set #4. The experimental results showed that the proposed method achieved 99.38%, 100% and 100% accuracy for early blight, healthy and late blight, respectively. It also attained 99.75% average accuracy on the PLD dataset, as shown in Table 7 and Figure 6a. The complete training, validation accuracy and losses in each epoch are depicted in Figure 7, showing the proposed PDDCNN method's overall performance on the PLD dataset using the data augmentation techniques applied to the training set. The results showed that the proposed method achieved excellent identification rates on the PLD dataset using the data augmentation techniques applied to the training set.

**Table 7.** Classification accuracies, precision, recall and F1-Score of the proposed PDDCNN model on the PLD dataset.

| | Performance Measures | Early Blight | Healthy | Late Blight | Average |
|---|---|---|---|---|---|
| **With Data Augmentation** | Accuracy | 99.38% | 100% | 100% | 99.75% |
| | Precision | 100% | 99% | 100% | - |
| | Recall | 99% | 100% | 100% | - |
| | F1-Score | 100% | 100% | 99% | - |
| **Without Data Augmentation** | Accuracy | 93.87% | 84.47% | 92.25% | 91.15% |
| | Precision | 87% | 94% | 90% | - |
| | Recall | 92% | 85% | 88% | - |
| | F1-Score | 96% | 92% | 94% | - |

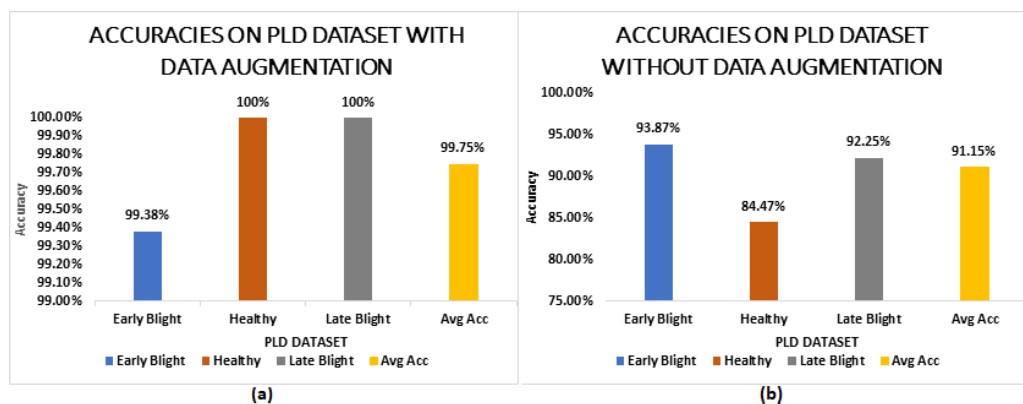

**Figure 6.** (**a**) Accuracies graph of PDDCNN on PLD with data augmentation. (**b**) Accuracies graph of PDDCNN on PLD without data augmentation.

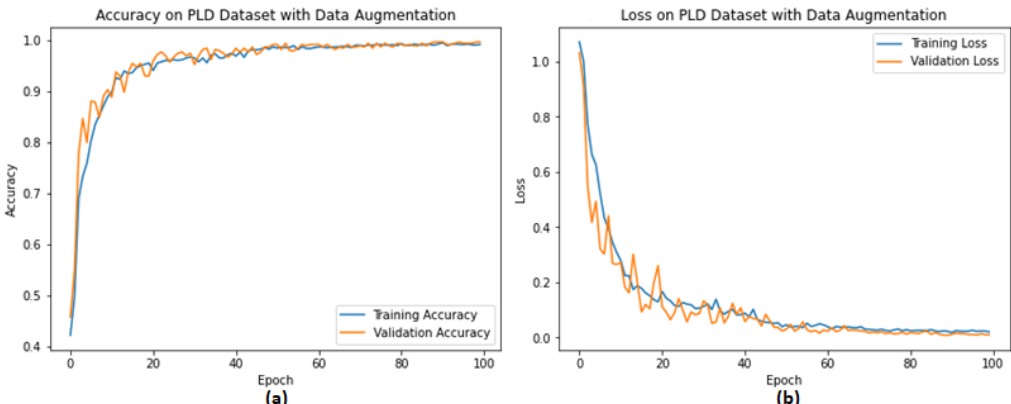

**Figure 7.** (**a**) Accuracy graph of PDDCNN on PLD using data augmentation. (**b**) Loss graph of PDDCNN on PLD using data augmentation.

The confusion matrix is a valuable Machine Learning method that calculates the precision, recall, accuracy and AUC-ROC curve. A confusion matrix was utilised to estimate the classification accuracy of a model visually. In the confusion matrix, correct predictions were exhibited diagonally, and incorrect predictions were shown off-diagonally. It represented the higher classification accuracy of the PDDCNN of the corresponding class in a dark colour, and a lighter colour meant the misclassified samples. The results showed that the proposed PDDCNN model performed significantly when data augmentation techniques were applied to the PLD dataset, as shown in Table 7. Figure 8a demonstrated that PDDCNN achieved 100%, 99% 100% precision scores on early blight, healthy and late blight and attained 99%, 100% 100% recall scores on early blight, healthy and late blight. The proposed PDDCNN model also achieved 99%, 100% 100% F1-scores on early blight, healthy and late blight using data augmentation techniques on the PLD dataset. The results showed excellent performance on all the classes of the PLD dataset. Figure 9a demonstrated the proposed PDDCNN method's confusion matrix. It showed that the proposed PDDCNN model correctly identified 162 early blight diseased images out of 163 images. All the healthy leaves (102) and all the early blight leaves (142) were also precisely classified. The overall classification accuracy of the proposed PDDCNN model was 99.75%, and the misclassification accuracy of the test set on all classes was 0.25%, which showed the proposed PDDCNN model's generalisation.

The proposed method's performance was measured using the ROC curve shown in Figure 10a. The light blue colour indicates the early blight. The orange colour represents the healthy class. The green colour represents the late blight class, and the blue colour shows the random guessing. All classes that showed a larger area (almost 100%) under the

curve present excellent classification performance of the PDDCNN model on the validation and test set.

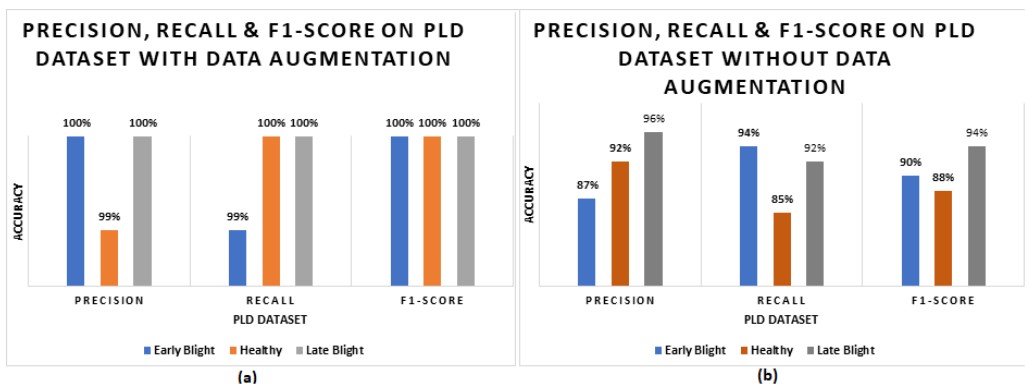

**Figure 8.** (**a**) PDDCNN precision, recall and F1-Score on the PLD dataset with data augmentation. (**b**) PDDCNN precision, recall and F1-Score on the PLD dataset without data augmentation.

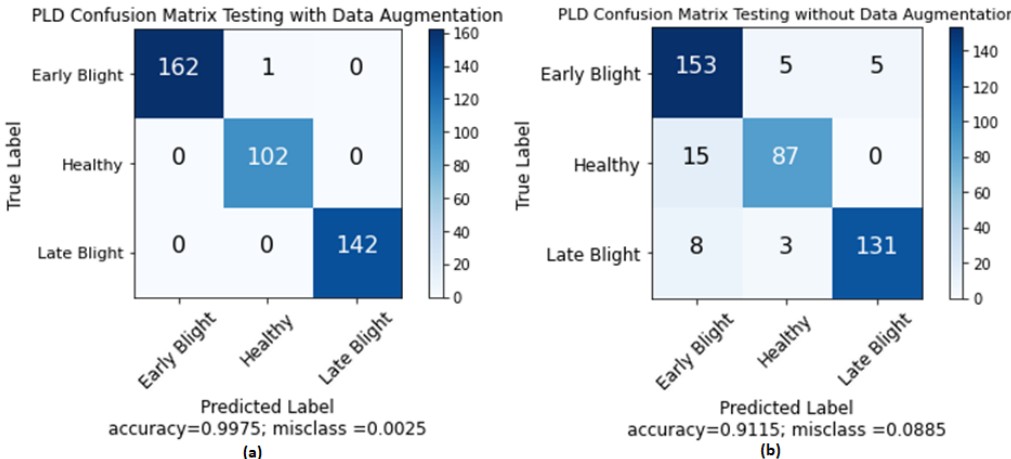

**Figure 9.** (**a**) The confusion matrix of PDDCNN on PLD with augmentation. (**b**) The confusion matrix of PDDCNN on PLD without augmentation.

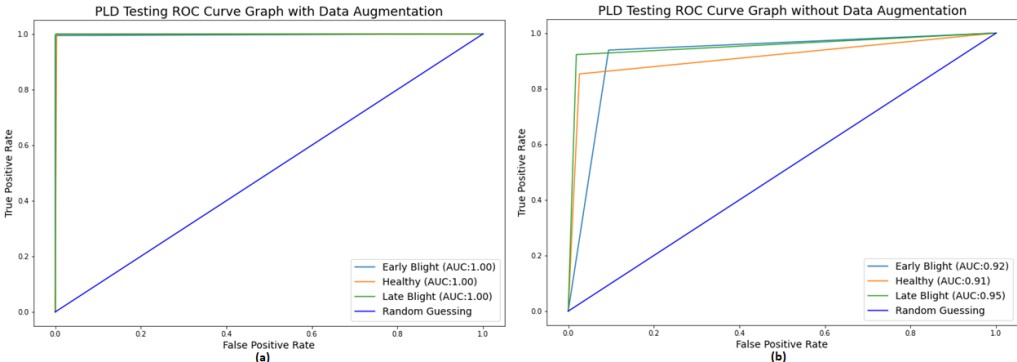

**Figure 10.** (**a**) ROC curve on the PLD dataset with augmentation techniques. (**b**) ROC curve on the PLD dataset without augmentation techniques.

All the evaluation measures, such as accuracy, precision, recall, F1-score and ROC curve, depicted the proposed method's excellent performance on the PLD dataset using the data augmentation techniques applied to the training set.

The proposed PDDCNN model performance was evaluated without applying the data augmentation techniques on the PLD dataset's training set in the second experiment. The same model and parameters were used as in the first experiment, but the number of epochs was reduced to 20. The proposed method achieved 93.87%, 84.47% and 92.25%

accuracy for early blight, healthy and late blight, respectively, and gained 91.1% average accuracy on the PLD dataset, as shown in Table 7 and Figure 6b. The complete training and validation accuracy and losses in each epoch were shown in Figure 11. The proposed method achieved lower identification rates without using data augmentation techniques compared to using data augmentation techniques applied to the training set. Therefore, for better classification accuracy, it should train on a large-scale dataset.

The proposed PDDCNN method's performance was further examined by calculating the precision, recall and F1-score on each class's validation set and testing set. The proposed PDDCNN model did not achieve better results when data augmentation techniques were not applied to the PLD's training set, as shown in Table 7 and shown in Figure 11b. On the PLD dataset, the PDDCNN achieved 87%, 94% and 90% precision on early blight, healthy and late blight, respectively, 92%, 85% and 88% recall for early blight, healthy and late blight, respectively, and it achieved 96%, 92% and 94% F1-scores on early blight, healthy and late blight. The results showed lower performance on all the classes than when data augmentation was applied to the PLD dataset training set. The reason for lower performance was that the model was trained on a limited dataset.

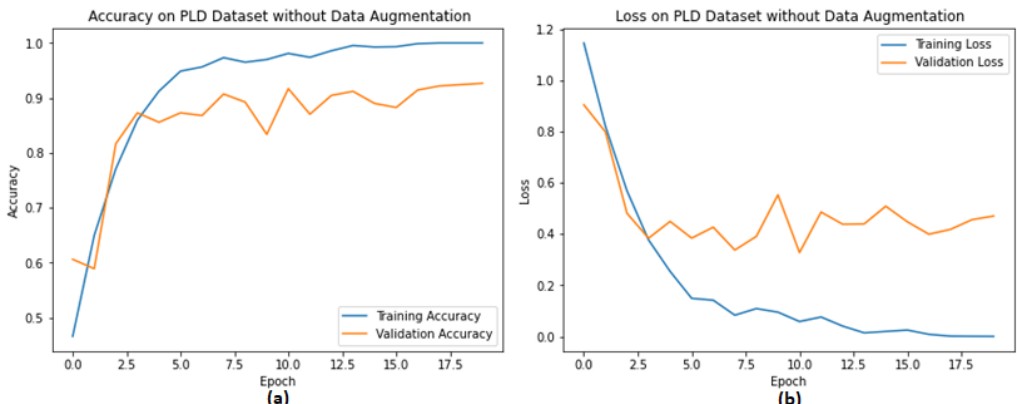

**Figure 11.** (**a**) Accuracy graph of PDDCNN on PLD without data augmentation. (**b**) Loss graph of PDDCNN on PLD without data augmentation.

The confusion matrix of the proposed PDDCNN method test set was also shown in Figure 9b. It was observed that the proposed PDDCNN model correctly identified the 153 early blight leaves out of 163 leaves, 87 healthy leaves out of 102 leaves were precisely identified, and 131 late blight leaf images out of 142 correctly predicted. The incorrect classification accuracy of the proposed method was 8.85%, which showed a lower performance of the proposed PDDCNN model compared to using data augmentation techniques. It meant that for training the proposed PDDCNN model, a massive number of images were required to improve the efficiency of the proposed PDDCNN model.

The ROC curve was calculated to evaluate the disease classification performance of the PDDCNN model depicted in Figure 10b. The blue colour indicates the early blight; the orange colour represents the healthy; the green colour represents the late blight; the blue colour indicates random guessing. The ROC curve graph showed that the early blight had 92%, healthy 91% and late blight with a 95% area under the curve, representing the good classification performance under the curve on the PLD dataset without data augmentation techniques applied to the training set of the proposed PDDCNN model.

All the evaluation measures demonstrated the proposed method's lower performance without applying the data augmentation techniques on the PLD dataset. It inferred that the proposed method required a massive amount of data for training. Less data produced the problem of overfitting. The overfitting could be eliminated by enhancing the dataset with the help of different data augmentation techniques or increase the dataset to millions of images.

### 4.2. The Proposed PDDCNN Model Performance on the PlantVillage Dataset

To generalise the proposed model, it trained on datasets other than the PLD dataset. For such purpose, we assessed the PDDCNN method performance by using the potato leaves from the publicly available dataset PlantVillage [23]. The database consisted of JPG colour images with 256 × 256 dimensions. It had 38 classes of diseased and healthy leaves of 14 plants. The focus of the research was on the potato crop. Therefore, 1000 leaves for late blight, 1000 leaves for early blight and 150 images of healthy leaves were selected for the experiment. Then the dataset was divided into 80%, 10% 10% ratios for train, validation and test sets. The training set consisted of 800 images of early blight, 125 images of healthy leaves and 800 images of late blight class; for the validation set, 100, 100 and 14 images of early blight, late blight and healthy, respectively. The test set contained 100, 100 and 13 images of early blight, late blight and healthy.

Two experiments were conducted on the proposed PDDCNN method using the PlantVillage dataset. In the first experiment, we applied the data augmentation techniques to the training set. The second experiment did not involve the data augmentation techniques to the training set of the PlantVillage dataset. The same model and parameters were used in the first experiment that were used in the previous section using the data augmentation techniques applied to the training set of the PlantVillage dataset. The proposed method's generalisation was checked by using training, validation and testing. The experiment results demonstrated that the proposed method achieved 99%, 92.31% and 95% accuracies for early blight, healthy and late blight classes, respectively. The proposed method achieved 96.71% average accuracy on the PlantVillage dataset depicted in Table 8 and Figure 12a. The complete training and validation accuracies and losses of each epoch were shown in Figure 13. The results showed that the proposed method achieved excellent identification rates when the data augmentation techniques applied the training set of the PlantVillage dataset, which indicated the proposed PDDCNN model's generalisation.

**Table 8.** Classification accuracies, precision, recall & F1-score of the proposed PDDCNN model on the PlantVillage dataset.

| | Performance Measures | Early Blight | Healthy | Late Blight | Average |
|---|---|---|---|---|---|
| **With Data Augmentation** | **Accuracy** | 99% | 92.31% | 95% | 96.71% |
| | **Precision** | 95% | 100% | 98% | - |
| | **Recall** | 99% | 92% | 95% | - |
| | **F1-Score** | 97% | 96% | 96% | - |
| **Without Data Augmentation** | **Accuracy** | 91% | 100% | 96% | 93.90% |
| | **Precision** | 97% | 93% | 91% | - |
| | **Recall** | 91% | 100% | 96% | - |
| | **F1-Score** | 94% | 96% | 94% | - |

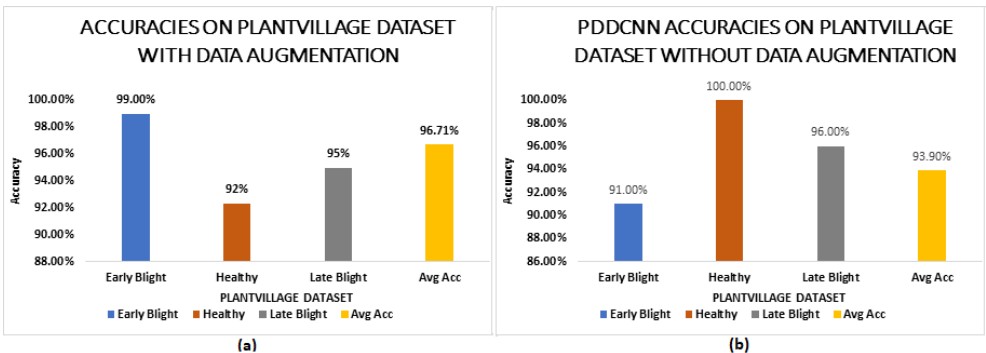

**Figure 12.** (**a**) Accuracies graph of PDDCNN on PlantVillage with data augmentation. (**b**) Accuracies graph of PDDCNN on PlantVillage without data augmentation.

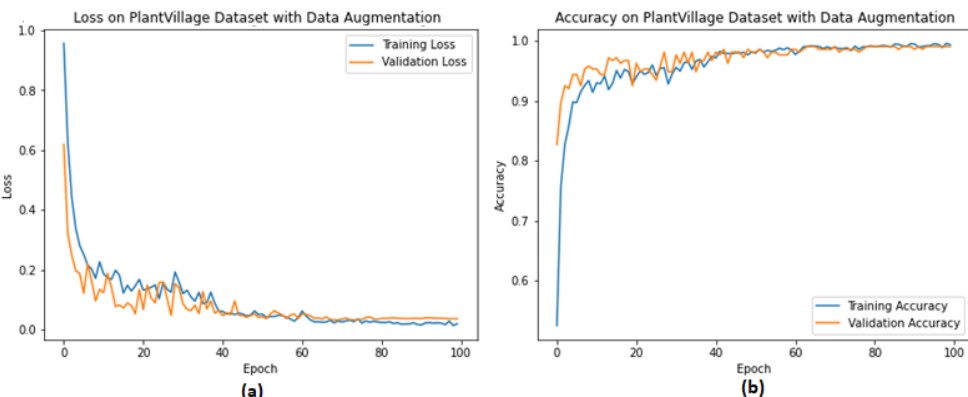

**Figure 13.** (**a**) Accuracy graph of PDDCNN on PlantVillage using data augmentation. (**b**) Loss graph of PDDCNN on PlantVillage using data augmentation.

The proposed method is further examined by calculating the test set's precision, recall and F1-score. It also observed that the proposed PDDCNN model's performance achieved excellent results using the data augmentation techniques applied to the training set of the PlantVillage dataset, as exhibited in Table 8 and Figure 14a. On the PlantVillage dataset, the PDDCNN achieved 95%, 100% and 98% precision on early blight, healthy and late blight, respectively. It attained 99%, 92% and 95% recall for early blight, healthy and late blight classes and 97%, 96% and 96% F1-scores on early blight, healthy and late blight. The results presented an excellent performance on all the classes of the PlantVillage dataset using data augmentation techniques.

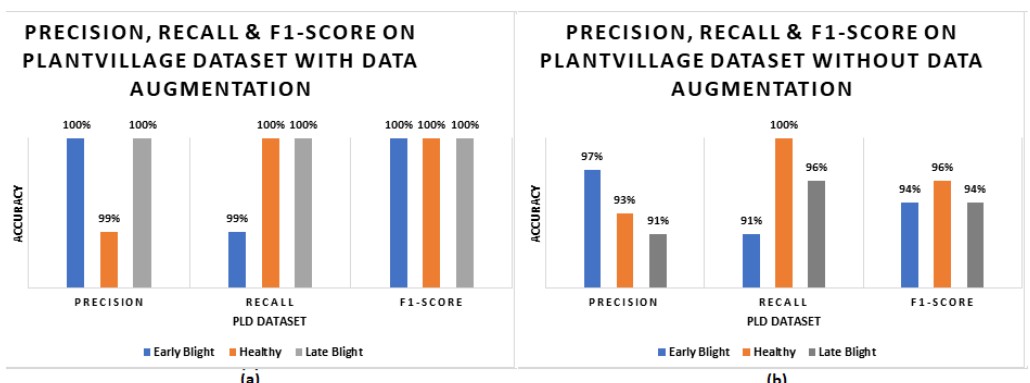

**Figure 14.** (**a**) PDDCNN precision, recall and F1-Score on PlantVillage with data augmentation. (**b**) PDDCNN precision, recall and F1-Score on PlantVillage without data augmentation.

The confusion matrix of the PlantVillage dataset with data augmentation techniques applied to the training set is also depicted in Figure 15a. The proposed PDDCNN model correctly identified 99 early blight diseased images out of 100 images, 12 healthy leaves out of 13 leaves, and 95 early blight leaves out of 100 leaf images. The overall classification accuracy of the proposed PDDCNN model was 96.71% on the PlantVillage dataset with data augmentation techniques applied to the training set. The proposed model's misclassification ratio was 3.29% on all classes when data augmentation techniques were applied to the training set on the PlantVillage dataset. The results confirmed that the proposed PDDCNN method achieved excellent prediction accuracy using data augmentation techniques applied to a training set of both the PLD and PlantVillage dataset.

ROC was also measured to confirm the proposed method's performance on the PlantVillage dataset using data augmentation techniques, as presented in Figure 16a. The light blue colour indicates the early blight. The orange colour denotes a healthy class. The green colour represents the late blight class, and the blue colour represents the random guessing. The ROC curve graph showed that the early blight achieved 97%,

healthy achieved 96%, and late blight achieved 97% area under the curve, representing good classification performance under the curve on the PlantVillage dataset using data augmentation techniques applied to the training set of the proposed PDDCNN model. All the evaluation measures showed that the proposed method achieved a lower accuracy on the PlantVillage dataset than the PLD dataset. The latter had more data for training the proposed model and, therefore, possessing more accuracy than the PlantVillage dataset.

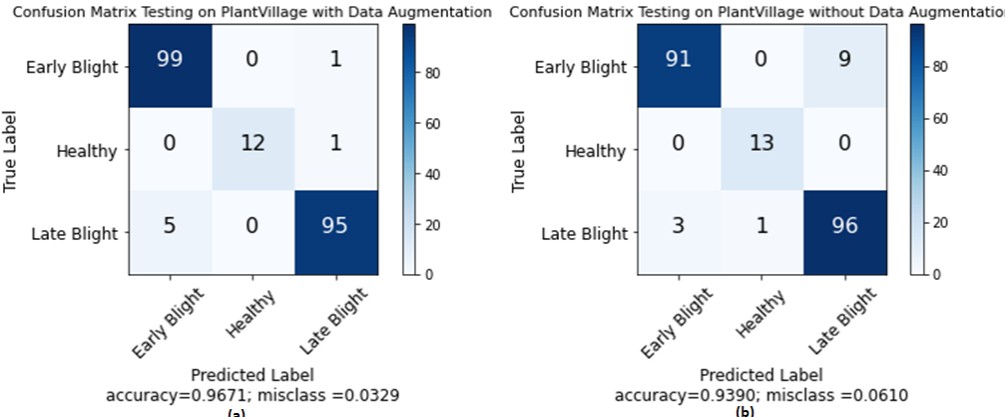

**Figure 15.** (**a**) PDDCNN confusion matrix on PlantVillage with augmentation. (**b**) PDDCNN confusion matrix on PlantVillage without augmentation.

The second experiment evaluated the proposed PDDCNN model performance without data augmentation techniques applied to the PlantVillage dataset's training set. It used the same previous experimental setup but reduced the epochs to 20. The experiment results showed that the proposed method achieved 91%, 100% and 96% accuracy for early blight, healthy and late blight, respectively. The proposed method attained 93.90% average accuracy on the PlantVillage dataset without using data augmentation techniques applied to the training set, as shown in Table 8 and Figure 12b. The complete training and validation accuracies and losses are shown in Figure 17, depicting the proposed PDDCNN method's overall performance on the PlantVillage dataset without data augmentation techniques applied to the training set.

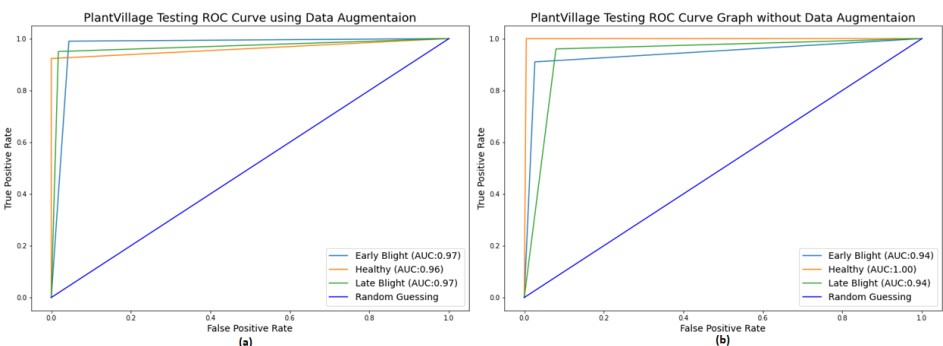

**Figure 16.** (**a**) PDDCNN ROC Curve on PlantVillage withaAugmentation. (**b**) PDDCNN ROC Curve on PlantVillage without augmentation.

The proposed method's performance was further verified by calculating the precision, recall, and F1-score, as shown in Table 8 and Figure 14b. On the PlantVillage dataset, the proposed method achieved 87%, 94% and 90% precision on early blight, healthy and late blight, respectively. It achieved 92%, 85% and 88% recall for early blight, healthy and late blight, respectively, and 96%, 92% and 94% F1-scores on early blight, healthy and late blight. The results showed lower performance on all the classes compared to when data augmentation was applied to the training set of the PlantVillage dataset. The confusion

matrix of the proposed PDDCNN method on the test set without using data augmentation techniques was shown in Figure 15b. The proposed model correctly identified the 91 early blight leaves out of 100 leaves. All healthy leaves (13) were precisely analysed, and 96 late blight leaf images out of 100 were also correctly predicted. The overall classification of accuracy was 93.90% on the test set, and the misclassification accuracy of the proposed method was 6.10%, which presented lower classification accuracy of the proposed model on the PlantVillage dataset without using the data augmentation techniques as compared to using data augmentation techniques.

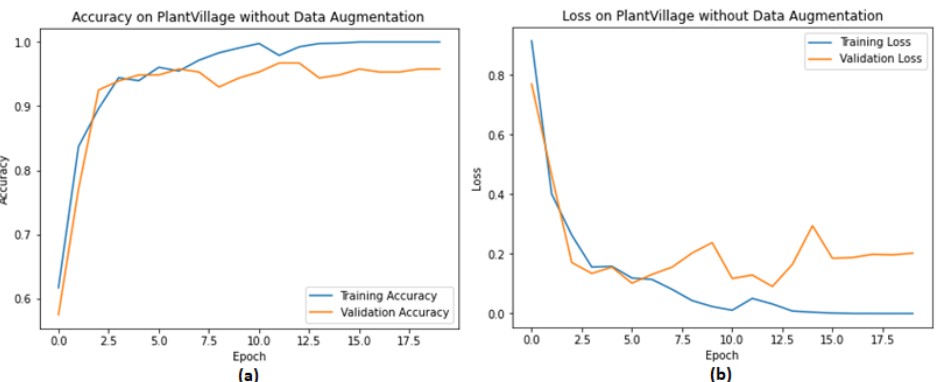

**Figure 17.** (**a**) Accuracy graph of PDDCNN on PlantVillage without data augmentation. (**b**) Loss graph of PDDCNN on PlantVillage without data augmentation.

Another evaluation measure, the ROC curve, was used to assess the proposed method's performance on the PlantVillage dataset without using data augmentation techniques, as shown in Figure 16b. The light blue colour indicates the early blight, and the orange colour represents the healthy class. The green colour denotes the late blight class, and the blue colour denotes random guessing. The ROC curve graph showed that the early blight had 94%, healthy had 100% and late blight had 94% area under the curve.

All results depicted the proposed PDDCNN method achieved less accurate performance when data augmentation techniques were not applied to the training set of the PlanVillage dataset. The reason is that the proposed method had less data to train. The deep learning method needed a massive amount of data to train. Therefore, to enhance the performance of the proposed method, the dataset should be very large.

### 4.3. Cross Dataset Performance

The performance of the proposed method was assessed by conducting two experiments on a cross dataset. In both experiments, using the same setup as in the first experiment, we performed training on the PlantVillage dataset and, in contrast, performed testing on the PLD dataset. In another experiment, we conducted training on the PLD dataset while we performed testing on the PlantVillage dataset. In the first experiment, the proposed method achieved 48.89% accuracy and, in the second experiment, 86.38%, as shown in Table 9 and Figure 18. The result confirmed the claim that plant species and diseases vary from region to region because of the different varieties, global warming and various environmental factors. There was a need for research in Pakistani crop disease detection.

**Table 9.** Classification accuracies of the proposed PDDCNN model training on PlantVillage and testing on PLD and training on PLD and testing on the PlantVillage dataset.

| Training Dataset | Testing Dataset | Early Blight Accuracy | Healthy Accuracy | Late Blight Accuracy | Total Testing Images | Overall Accuracy |
|---|---|---|---|---|---|---|
| PlantVillage | PLD | 95.71% | 08.82% | 23.94% | 807 | 48.89% |
| PLD | PlantVillage | 92.00% | 100% | 79.00% | 213 | 86.38% |

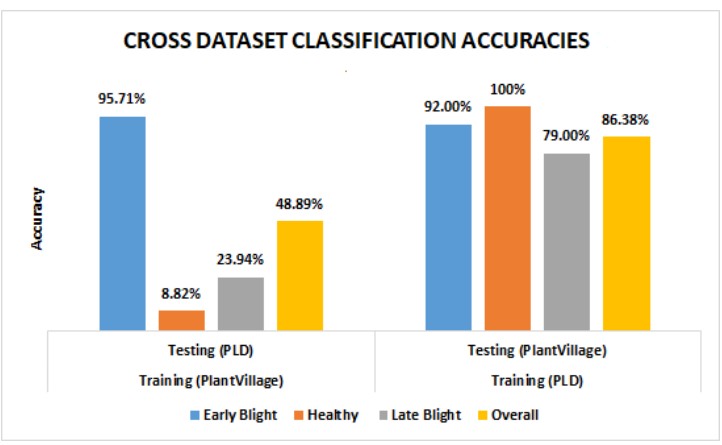

**Figure 18.** PDDCNN cross dataset classification accuracies.

### 4.4. Accuracies Comparison of Proposed Method with State-of-the-Art Methods

To evaluate the performance of the proposed PDDCNN model, we performed transferred learning on VGG16 [31], InceptionResNetV2 [45], DenseNet_121 [46], DenseNet169 [46] and Xception [47] models with the proposed PDDCNN model on the PLD dataset. For such purpose, the experiments had the same setup and same data augmentation techniques. Table 10 shows the accuracy achieved by the state-of-the-art deep learning techniques. The VGG16 model achieved 40.05% accuracy; InceptionResNetV2 accomplished 99.26% accuracy; DenseNet_121 model gained 99.26% accuracy; DenseNet169 model carried off 99.53% accuracy; Xception model reached 99.26% accuracy; the proposed PDDCNN model acquired 99.75% accuracy on the PLD dataset, as shown in Table 10. The results exhibited that the proposed PDDCNN model achieved the highest accuracy (99.75%), and VGG16 achieved the worst accuracy (40.05%). The proposed PDDCNN method had less training parameters than the VGG16, InceptionResNetV2, DenseNet_121, DenseNet169 and Xception. The state-of-the-art models had a large number of trainable parameters as compared to the proposed PDDCNN model, i.e., 14,716,227, 20,867,627, 7,040,579, 12,647,875 and 20,867,627 parameters for VGG16, Inception ResNetV2, DenseNet_121, DenseNet169 and the Xception model, respectively, as presented in Table 10. In contrast, the proposed PDDCNN model had only 8,578,611 parameters, and DenseNet_121 model had the lowest with 7,040,579 parameters, as shown in Table 10, which saved a lot of computational costs and needed less time to train the model than the state-of-the-art models except the DenseNet_121 model. The proposed PDDCNN model had fewer convolutional layers; fewer layers meant fewer parameters and less computational cost.

**Table 10.** Comparison with state-of-the-art techniques.

| Model | Total Parameters | Accuracy |
|---|---|---|
| VGG16 [31] | 14,716,227 | 40.05% |
| InceptionResNetV2 [45] | 20,867,627 | 99.26% |
| DenseNet_121 [46] | 7,040,579 | 99.26% |
| DenseNet169 [46] | 12,647,875 | 99.53% |
| Xception [47] | 20,867,627 | 99.26% |
| PDDCNN | 8,578,611 | 99.75% |

### 4.5. Accuracies Comparison of Proposed Method with Existing Studies

To represent the proposed approach's generalisation, we had compared the proposed technique's performance with the state-of-the-art methods, as exhibited in Table 10. It observed that the proposed deep learning model performed significantly well when compared to state-of-the-art techniques. There was a slight difference in the accuracy of the proposed approach and the state-of-the-art techniques. The proposed method's performance was compared to the potato leaf disease detection existing techniques from the

literature. The results showed that the proposed method achieved the highest accuracy (99.75%) compared to existing studies, as shown in Table 11. The proposed PDDCNN model outperformed the existing studies as Tiwari et al. [37] achieved 97.80% accuracy, but pre-trained models were employed having a large number of trainable parameters, i.e., 143,667,240. The PlantVillage dataset was used to detect potato disease detection, which had imbalanced classes and a smaller number of images. Khalifa et al. [33] reported 98.00% accuracy possessing 14 layers architecture with higher computational cost. The PlantVillage dataset was utilised, containing imbalanced classes and a smaller number of images. Lee et al. [38] performed 99.00% accuracy, but it had 10,089,219 trainable parameters and used the PlanVillage dataset, which possesses imbalanced classes and a smaller number of parameters. The proposed PDDCNN model dominated the existing techniques, thus achieving 99.75% accuracy with fewer parameters, i.e., 8,578,611, leading to a lower computational cost and the highest accuracy compared to existing models.

**Table 11.** Comparison with existing studies.

| Existing Study | Total Parameters | Accuracy |
|---|---|---|
| Rozaqi and Sunyoto [34] | 6,812,995 | 92.00% |
| Islam et al. [39] | - | 95.00% |
| Sanjeev et al. [35] | - | 96.50% |
| Barman et al. [36] | 16,407,395 | 96.98% |
| Tiwari et al. [37] | 143,667,240 | 97.80% |
| Khalifa et al. [33] | - | 98.00% |
| Lee et al. [38] | 10,089,219 | 99.00% |
| PDDCNN | 8,578,611 | 99.75% |

## 5. Conclusions and Future Work

Deep learning techniques perform significantly in plant leaf disease detection to improve crop productivity and quality by controlling the biotic variables that cause severe crop yield losses. In this study, a fast and straightforward multi-level deep learning model for potato leaf disease recognition was proposed to classify the potato leaves diseases. It extracted the potato leaves from the potato plant image at the first level using the YOLOv5 image segmentation technique, then developed a novel potato leaf disease detection convolutional neural network (PDDCNN) at the second level to classify early blight and late blight potato diseases from potato leaf images. At the same time, it considered the effect of the environmental factors on potato leaf diseases. The proposed PDDCNN method performed significantly well on the potato leaf images collected from Central Punjab, Pakistan. Experimental studies were conducted on two different datasets, PlantVillage and PLD, with and without augmentation. The performance of the proposed PDDCNN techniques was also evaluated in the cross dataset, where the proposed approach outperformed the other methods. The proposed technique's performance was compared with the state-of-the-art techniques, and existing studies were used for potato leaf disease detection. The state-of-the-art techniques and existing techniques had a high false rate in detecting the potato leaf disease on the PLD dataset, which strengthened the effect of environmental factors and disease symptoms variation in the PlantVillage dataset and PLD dataset. The proposed method was trained on the PLD dataset with and without data augmentation techniques, thus achieving 99.75% accuracy, high precision, recall, F1-score and roc curve on the PLD dataset. It had a minimal number of parameters and was simpler than the state-of-the-art methods, saving a substantial computational cost and speed.

In future, such research would be extended to multiple diseases detection on a single leaf and to localise the diseases, disease severity estimation, enhance the PLD dataset, develop IoT-based real-time monitoring system, develop a website and launch a mobile application.

**Author Contributions:** J.R. proposed the research conceptualisation, methodology and programming. The technical and theoretical framework is prepared by I.K. and G.A. Dataset creation have been performed by J.R. and S.H.A. The technical review and improvement have been performed by M.A.A., K.M. and S.H.A. The overall technical support, guidance and supervision is done by I.K. and G.A. The editing and proofreading was done by J.R. and G.A. All authors have read and agreed to the published version of the manuscript.

**Funding:** This research received no external funding.

**Conflicts of Interest:** The authors declare no conflict of interest.

## Abbreviations

The following abbreviations are used in this manuscript:

| | |
|---|---|
| PDDCNN | Potato Leaf Disease Detection using Convolutional Neural Network |
| FAO | Food and Agriculture Organisation Report |
| AI | Artificial Intelligence |
| ML | Machine Learning |
| CV | Computer Vision |
| CNN | Convolutional Neural Network |
| ANN | Artificial Neural Network |
| PLD | Potato Leaf Dataset |
| RCNN | Region Based Convolutional Neural Networks |
| VGG | Visual Geometry Group |
| ResNet | residual neural network |
| FFNN | Feed-Forward Neural Network |
| SBCNN | Self-Build CNN |
| KNN | K-Nearest Neighbor |
| SVM | Support Vector Machine |
| ROI | Region of Interest |
| GPU | Graphics Processing Unit |
| ReLu | Rectified Linear Unit |
| ROC | Receiver Operating characteristic |
| TP | True Positive |
| FP | False Positive |
| FN | False Negative |
| FP | False Positive |
| TPR | True Positive Rate |
| FPR | False Positive Rate |
| AUC | Area under the ROC Curve |

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
