# Peer review of "Multi-Level Deep Learning Model for Potato Leaf Disease Recognition"

_electronics, doi:10.3390/electronics10172064_

Round 1

Reviewer 1 Report

Thank you for your work on this important topic!  I personally crop a variety of all-purpose and waxy potatoes in both open field and integrated forest systems, and very much appreciate the challenges you are helping address in this research :)  

Typos and minor edits:

  1. Please review nouns for pluralization and verbs for conjugation.  There are many mismatches.
  2. L81: "The authors rst applied" -> "The authors first applied"
  3. L185: "The ow chart" -> "The flow chart"
  4. L484: "and miss-classication" -> "and misclassification"

Suggestions:

  1. As you indicate, there are many varieties of potatoes grown around the world, and many of these potatoes exhibit dramatically different leaf morphology.  Can you please indicate which varieties are present in the data collected from Punjab?
  2. Can you please describe the strategy used for growing these potatoes?  For example, does the sample include potatoes that were planted in a row crop field?  Were they hilled up mechanically?  Was the field rotated across brassica/etc. or permanently cropped on solanaceae?  Some of this information would be useful to help diversify training data and interpret results across PlantVillage/etc. 
  3. A full confusion matrix or report including F1/precision/recall for all classes might be more useful than the accuracy table as presented.
  4. The publication and model would likely receive more citations and use if they were hosted on a resource like Hugging Face, GitHub, or Zenodo.
  5. Have the authors considered packaging their model as a Python library or providing a mobile app to integrate data collection with a hosted web service (similar to PlantNet mobile apps)?

Author Response

Please the attachement.

Reviewer 2 Report

I want to support this paper. It my opinion paper is well prepared, methods are described well and results are supported by evidence. 

Author Response

Respected Reviewer,

Thanks for appreciation. we have corrected the typos and gramartic mistakes throughout the article.

Regards

Reviewer 3 Report

The paper presents a deep learning technique using a convolutional neural network to detect the early blight and late blight potato diseases from
 potato leaf images. 

The abstract is hard to read. It should contain some main summary items and not a partial introduction.

What are the contributions of the paper?

The architecture is not new. The novelty of the article is not compared to the state-of-the-art sufficiently in the introduction section. It is essential to demonstrate the problem at stake and the drawbacks of existing studies when proposing the approach.  

The dataset needs to explain clearly. How do you collect the samples?

what type of data augmentation did u use? Pls compare different methods like mixup.

The number of samples/images seen too low.

Round 2

Reviewer 3 Report

I have no further comment.